# Comparative physiological and transcriptomic analyses reveal the mechanisms of $CO_2$ enrichment in promoting the growth and quality in *Lactuca sativa*

**Hongxia Song**[1], **Peiqi Wu**[1], **Xiaonan Lu**[1], **Bei Wang**[1], **Tianyue Song**[1], **Qiang Lu**[1], **Meilan Li**[1], **Xiaoyong Xu**[1,2]*

**1** College of Horticulture, Shanxi Agricultural University, Taigu, Shanxi, China, **2** Hainan Yazhou Bay Seed Lab, Sanya, Hainan, China

* xuxy@yazhoulab.com

**Data Availability Statement:** All relevant data are within the article and its Supporting Information files.

## Abstract

The increase in the concentration of $CO_2$ in the atmosphere has attracted widespread attention. To explore the effect of elevated $CO_2$ on lettuce growth and better understand the mechanism of elevated $CO_2$ in lettuce cultivation, 3 kinds of lettuce with 4 real leaves were selected and planted in a solar greenhouse. One week later, $CO_2$ was applied from 8:00 a. m. to 10:00 a.m. on sunny days for 30 days. The results showed that the growth potential of lettuce was enhanced under $CO_2$ enrichment. The content of vitamin C and chlorophyll in the three lettuce varieties increased, and the content of nitrate nitrogen decreased. The light saturation point and net photosynthetic rate of leaves increased, and the light compensation point decreased. Transcriptome analysis showed that there were 217 differentially expressed genes (DEGs) shared by the three varieties, among which 166 were upregulated, 44 were downregulated, and 7 DEGs were inconsistent in the three materials. Kyoto Encyclopedia of Genes and Genomes (KEGG) analysis showed that these DEGs involved mainly the ethylene signaling pathway, jasmonic acid signaling pathway, porphyrin and chlorophyll metabolism pathway, starch and sucrose metabolism pathway, etc. Forty-one DEGs in response to $CO_2$ enrichment were screened out by Gene Ontology (GO) analysis, and the biological processes involved were consistent with KEGG analysis. which suggested that the growth and nutritional quality of lettuce could be improved by increasing the enzyme activity and gene expression levels of photosynthesis, hormone signaling and carbohydrate metabolism. The results laid a theoretical foundation for lettuce cultivation in solar greenhouses and the application of $CO_2$ fertilization technology.

## 1. Introduction

$CO_2$ is an important raw material for plant photosynthesis, and an increase in its concentration can promote the synthesis of photosynthetic products and affect plant growth and physiological and biochemical metabolic processes, the antioxidant system and secondary metabolic

**Funding:** This work was supported by the Shanxi Province Key Research and Development Program Key Projects under Grant (201703D211001-04-01, 201903D211011), The role of the funders was the study design.

**Competing interests:** The authors have declared that no competing interests exist.

processes. Therefore, the effect of elevated $CO_2$ in the atmosphere on plant growth has attracted extensive attention from researchers [1, 2]. Greenhouse vegetables are usually in a relatively closed environment, and $CO_2$ deficit is an important limiting factor of yield decline. Therefore, it is of great practical value to study $CO_2$ enrichment.

C3 plants are more sensitive to $CO_2$ concentration than C4 crops, therefore, C3 plants will increase their biomass due to the increase of carbon dioxide concentration. The physiological changes caused by high $CO_2$ concentrations were reflected mainly in growth and development, nutritional quality, photosynthetic response, etc. Palit et al. [3]. found that an increased $CO_2$ concentration changed the shoot and root length, nodulation rate, chlorophyll content and nitrogen balance index of chickpeas significantly. Hikosaka et al. [4]. found that the increase in nitrogen content in seeds led to an increase in seed quality per plant in $C_3$ plants under elevated $CO_2$. The increase in $CO_2$ concentration and temperature can promote the balanced development of source–sink organs and have a positive effect on potato yield and quality [5]. Becker and Kläring found that an increase in $CO_2$ concentration not only increased the yield of red leaf lettuce but also increased the concentration of flavonoid glycosides in leaves [6].

RNA-sequencing (RNA-seq) can comprehensively and immediately access full transcript information and be used to study the differences in gene expression in different environments. Xu et al. [7] found that after high-concentration $CO_2$ treatment, many genes in plants showed changes in the transcription level, and the expression levels of genes involved in the light response were all upregulated. Our early study in cucumbers analyzed the chlorophyll metabolic pathway under $CO_2$ enrichment and screened out 17 differentially expressed genes (DEGs) [8]. Zheng et al. found that 208 DEGs involved in photosynthesis and sugar synthesis responded to elevated $CO_2$ enrichment in tomato [9]. Xu et al. [10] found 169 DEGs in eggplant treated with high concentrations of $CO_2$, which are involved mainly in carbon metabolism, carbon fixation, chlorophyll and porphyrin metabolism and other pathways.

Lettuce (*Lactuca sativa*) is widely cultivated and annually supplied in China due to its high market demand and high economic benefit. Because of the lower requirement for temperature, it is also one of the main vegetables cultivated in winter and spring facilities in northern regions. The greenhouses are often not ventilated due to the low external temperature outside [11]; thus, the $CO_2$ in greenhouses is consumed by vegetables after sunrise and decreases rapidly to a very low level, which affects yield and quality. In this study, three different colors of lettuce were studied for their physiological changes in growth, nutritional quality, photosynthetic response and other aspects by applying exogenous $CO_2$ in a greenhouse. At the same time, RNA-seq was used to analyze the expression profile of elevated $CO_2$-related genes in lettuce, aiming to screen and analyze the key elevated $CO_2$-responsive genes and related metabolic pathways in lettuce and reveal the molecular mechanism of its response to elevated $CO_2$.

## 2. Materials and methods

### 2.1 Plant cultivation and $CO_2$ application

S6 (green), S16 (green and purple) and S24 (purple) lettuce were selected as the research material in this study (S1 Fig). The study was carried out in a solar greenhouse that was separated by a plastic film into control (approximately 400 $\mu mol \bullet mol^{-1}$) and carbon-enriched zones (800 $\pm 50$ $\mu mol \bullet mol^{-1}$). The seedlings were transplanted at the 4-true leaf stage using soil cultivation, and elevated $CO_2$ was applied after one week. $CO_2$ fumigation period was 8:00 a.m.-10:00 a.m, and no $CO_2$ was released on rainy or snowy days, and there were 30 d for $CO_2$ application in general. The $CO_2$ concentration in carbon-enriched zones was controlled by a 'Greenhouse $CO_2$ Automatic Control System' (Shengyan Electronic Scientific Technology Co., Ltd., Handan, Hebei, China). The plants were cultivated according to zhang [12]' method.

## 2.2 Determination of morphology, photosynthetic indices and nutritional quality

Morphological and photosynthetic indices were measured 25 days after $CO_2$ application, and morphological indices included mainly leaf length, leaf width, plant height and plant width. Leaf width refers to the widest distance perpendicular to the main vein. Plant height refers to the distance from the base of the plant to the highest point. Plant width refers to the maximum width that can be formed by the above-ground part. Six plants were used as one sample per treatment with three replicates.

Photosynthetic indicators were measured using a LI-6400 XT portable photosynthetic instrument (LI-COR Biosciences Inc., USA) at 9:00–11:30 am including Gs and intercellular $CO_2$ concentration. Built-in red and blue light sources were used for the measurement, and the temperature in the chamber was set to 25˚C. The light intensity was set as 1000 μmol•m$^{-2}$•s$^{-1}$. One leaf per plant was chosen for conducting the light curve, and the light intensity settings were 50, 100, 150, 200, 300, 400, 600, 800, 1000, 1200, 1400, 1600, 1800, 2000 and 2200 μmol• m$^{-2}$•s$^{-1}$, respectively, three healthy plant s were selected for each measurement.

Nutritional quality was measured 30 d after $CO_2$ treatment, and 10 plants with roughly the same growth vigor were mixed into a sample. Each treatment was repeated three times. Chlorophyll was extracted by the alcohol method [8], and the organic acid content was determined by the acid-base titration method in GB/T12456-2008 (2008). Vitamin C was determined by 2,6-dichlorophenol-indophenol titration [13], nitrate nitrogen was titrated by the salicylic acid method [14], and soluble solids were determined by the refractometer method (NY/T 2637–2014).

The comparison between elevated $CO_2$ and ambient $CO_2$ was done for the statistical significance for each variable (from plant growth, photosynthesis, lettuce quality) of each variety. No significant difference analysis was performed between varieties.

## 2.3 Transcriptome sampling and gene expression sequencing

After 25 days of $CO_2$ application, the functional leaves of three plants with uniform growth of each material were selected from the control area and the carbon-rich area, wrapped in aluminum foil paper, sealed in plastic bags, quickly frozen in liquid nitrogen, and then outsourced to Biomarker Technologies Co., Ltd. (Beijing, China). The samples in the control and treatment areas were labeled SCD6, SCD16, and SCD24 and SCF6, SCF16, and SCF24, respectively. There is one replicate used for RNA-seq from S6, S16, S24 varieties. In this study, transcriptome data of three cultivars were analyzed as three biological replicates.

Gene expression sequencing was performed by Biomarker Technologies Co., Ltd., Beijing, China. Raw reads from each sample were processed by removing rRNA and low-quality reads to obtain clean reads. The clean reads from each library were aligned to the lettuce (*Lactuca sativa* cv *Salinas* V8 Plus unmmaped sequences) (https://genomevolution.org/CoGe/GenomeInfo.pl?gid=35223) using TopHat2. DEGs between different samples were identified using EBSeq software. DEGs were identified for each lettuce variety separately. The sequencing steps, expression analysis and functional annotation methods applied by the company were the same as Sun et al. [15].

## 2.4 Screening of DEGs and metabolic pathways in response to carbon enrichment

A false discovery rate (FDR)<0.01 and fold change (FC)≥2 were used as the thresholds to screen DEGs. Volcano and Venn diagrams were drawn using the R package. Gene function

and pathway involved in were analyzed using the Gene ontology (GO) (http://www.geneontology.org/) and Kyoto Encyclopedia of Genes and Genomes (KEGG) (http://www.genome.jp/kegg/) databases. GO enrichment analysis of the DEGs was implemented by the GOseq R packages based Wallenius non-central hyper-geometric distribution. KO-Based Annotation System software test the statistical enrichment of DEGs in KEGG pathways. KEGG pathway annotation enrichment and GO enrichment analysis were performed for DEGs to screen genes and metabolic pathways responding to carbon enrichment, and a corrected P value < 0.05 was considered.

## 2.5 Candidate gene quantitative real-time polymerase chain reaction (qRT-PCR) validation

Eight carbon-rich response genes were randomly selected for qRT-PCR verification. *Ubiquitin* from lettuce was used as the reference gene, and the primer sequences are shown in S1 Table. The total RNA of each sample tissue was extracted using an RNA extraction kit (Tiangen, DP171221). cDNA was synthesized using the 1$^{st}$ Strand cDNA Synthesis Kit. qRT-PCR amplification was performed with SuperReal PreMix Color (SYBR Green) (Tiangen, R6332) on an ABI 7500 real-time PCR system. The procedure was as follows: 94˚C for 5 min and 40 cycles of 95˚C for 30 s, annealing and extension at 56˚C. Finally, the relative levels of target genes were calculated by the $2^{-\triangle\triangle Ct}$ method [16].

## 2.6 Statistical analysis

The data are presented as the means ± one standard deviation (SD) of three replicates. The statistical analyses were analyzed with one-way ANOVA and performed by the Statistical Analysis System (SAS, North Carolina, USA).

## 2.7 Data access

The transcriptome sequencing data from this study have been deposited in the National Center for Biotechnology Information Sequence Read Archive database, and are accessible through accession number PRJNA859388 (http://www.ncbi.nlm.nih.gov/bioproject/859388).

# 3. Results

## 3.1 Effects of elevated $CO_2$ on the leaf morphology of lettuce

Elevated $CO_2$ had a certain promotion effect on the aboveground growth (Table 1), especially the leaf length and plant height of the three varieties. There were some differences in the sensitivity of different varieties to $CO_2$ enrichment; the leaf width only increased significantly in S24, while the plant width only increased significantly in S6.

## 3.2 Effects of elevated $CO_2$ on the photosynthesis characteristics of lettuce

The light saturation point (LSP) increased significantly, and the light compensation point (LCP) decreased significantly in the three cultivars under elevated $CO_2$ conditions (Table 2). The maximum net photosynthetic rate (Pn) and intercellular $CO_2$ concentration increased. Stomatal conductance (Gs) and intercellular $CO_2$ concentration antagonized each other, and Gs decreased with increasing $CO_2$ application.

**Table 1. Effect of elevated $CO_2$ on the growth of lettuce.**

| | | Leaf length | Leaf width | Plant height | Plant width |
|---|---|---|---|---|---|
| | | (cm) | (cm) | (cm) | (cm) |
| S6 | Elevated $CO_2$ | 18.33±0.07 a | 10.13±0.07 a | 18.83±0.22 a | 18.83±0.17 a |
| | Ambient $CO_2$ | 14.90±0.29 b | 9.73±0.15 a | 15.17±0.17 b | 14.83±0.36 b |
| S16 | Elevated $CO_2$ | 14.67±0.27 a | 11.67±0.17 a | 15.33±0.44 a | 17.63±0.67 a |
| | Ambient $CO_2$ | 11.67±0.33 b | 10.33±0.08 a | 12.33±0.33 b | 17.17±0.76 a |
| S24 | Elevated $CO_2$ | 17.00±0.29 a | 9.23±0.38 a | 17.50±0.29 a | 20.17±0.17 a |
| | Ambient $CO_2$ | 15.00±0.16 b | 7.17±0.17 b | 16.00±0.50 b | 20.00±0.08 a |

Note: small letters in each table represent significant differences (P < 0.05). Labels in the figures and tables below are the same.

## 3.3 Effects of elevated $CO_2$ on the quality of lettuce

Leaf vitamin C, chlorophyll and soluble solids all increased under elevated $CO_2$ conditions in the three lettuce varieties (Table 3), while the organic acids and nitrate nitrogen significantly decreased, indicating that $CO_2$ enrichment promoted the improvement of lettuce quality.

## 3.4 Screening related to mechanic pathways responding to elevated $CO_2$

**3.4.1 Statistical analysis of the sequencing data and sequencing quality assent.** RNA-seq analysis was conducted on the leaves of three lettuce materials (S6, S16 and S24) in the control area and the carbon-rich area. After quality control of the original data, a total of 43.02 Gb of high-quality sequencing data were obtained, and the sequencing data of each sample were greater than 6.24 Gb. Statistical analysis of the sequencing data after quality control (S2 Table) showed that the percentage of GC content in each sample was 44.33%-45.56%, and the percentage of Q30 bases in each sample was 88.87%-90.42%. After removing low-quality reads, the comparison rates of clean reads and the reference genome of the three $CO_2$-treated samples (SCF6, SCFB16 and SCF24) were 82.71%, 76.53% and 78.83%, respectively. The comparison rates of the three samples (SCD6, SCDB16 and SCD24) with the reference genome were 81.16%, 65.14% and 78.96%, respectively. In addition, by comparing the annotated information with the lettuce genome, 1010 new unannotated genes were found after the removal of some of the coding short peptide chains (less than 50 amino acid residues) (S3 Table).

**3.4.2 Screening of differentially expressed genes responding to elevated $CO_2$.** FC≥2 and FDR<0.01 were used as the screening criteria for DEGs. The results showed that 1384 DEGs were screened in S6 in the control region and the carbon-rich region; among these

**Table 2. Effect of elevated $CO_2$ on photosynthesis in lettuce.**

| | | Pn max | Lsp | Lcp | AQE | Gs | Ci |
|---|---|---|---|---|---|---|---|
| | | $\mu mol \cdot m^{-2} \cdot s^{-1}$ | $\mu mol \cdot m^{-2} \cdot s^{-1}$ | $\mu mol \cdot m^{-2} \cdot s^{-1}$ | | $mol \cdot m^{-2} \cdot s^{-1}$ | $\mu mol \cdot mmol^{-1}$ |
| S6 | Elevated $CO_2$ | 38.24±2.07 a | 2673.60±25.52 a | 11.20±3.52 b | 0.14±0.001 b | 0.19±0.01 b | 479.54±23.01 a |
| | Ambient $CO_2$ | 21.85±1.98 b | 1273.63±13.18 b | 33.62±4.87 a | 0.13±0.003 b | 0.23±0.01 a | 262.29±12.92 b |
| S16 | Elevated $CO_2$ | 22.29±1.05 a | 2312.83±49.62 a | 11.21±1.38 b | 0.21±0.002 a | 0.17±0.01 b | 470.56±16.24 a |
| | Ambient $CO_2$ | 10.74±1.27 b | 1310.46±22.14 b | 28.94±3.30 a | 0.08±0.004 b | 0.25±0.02 a | 250.28±6.84 b |
| S24 | Elevated $CO_2$ | 40.96±2.15 a | 2844.87±49.77 a | 10.47±2.27 b | 0.12±0.001 a | 0.17±0.01 a | 517.84±18.34 a |
| | Ambient $CO_2$ | 23.42±0.52 b | 1347.35±180.24 b | 16.82±4.35 b | 0.11±0.002 a | 0.19±0.01 a | 284.73±4.03 b |

Note: Pn max: the maximum net photosynthetic rate, Lsp: light saturation point, Lcp: light compensation poin, AQE: apparent quantum efficiency, Gs: stomatal conductance, Ci: intercellular $CO_2$ concentration.

**Table 3. Effect of elevated $CO_2$ on the lettuce quality.**

| | | Vitamin C(mg$^*$g/FW) | Organic acid(mol/(g$^*10^3$) | Chlorophyll (mg$^*$/g FW) | Soluble solids(%) | Nitrate nitrogen (µgNO$_3$.N$^*$FW/g) |
|---|---|---|---|---|---|---|
| **S6** | **Elevated $CO_2$** | 1.07±0.24 a | 0.47±0.03 B | 0.36±0.02 a | 7.43±0.11 B | 672.62±11.83 B |
| | **Ambient $CO_2$** | 0.60±0.06 b | 0.70±0.10 A | 0.31±0.01 b | 7.40±0.20 A | 2587.30±30.33 A |
| **S16** | **Elevated $CO_2$** | 7.13±0.47 A | 0.35±0.02 a | 0.28±0.01 a | 6.63±0.26 A | 631.98±9.37 B |
| | **Ambient $CO_2$** | 5.33±0.45 B | 0.28±0.04 a | 0.26±0.01 a | 4.47±0.25 B | 753.97±9.95 A |
| **S24** | **Elevated $CO_2$** | 4.57±0.08 A | 0.43±0.09 a | 0.66±0.04 A | 5.47±0.13 A | 343.25±42.73 B |
| | **Ambient $CO_2$** | 2.27±0.08 B | 0.45±0.03 a | 0.55±0.07 B | 2.37±0.17 B | 569.44±29.43 A |

Note: Capital letters in each table represent extremely significant differences ($P < 0.01$). Difference analysis was not performed between varieties, but significant difference analysis was performed only between controls and treatments of the same variety.

DEGs, 872 were upregulated, and 512 were downregulated (Fig 1A). There were 1215 DEGs between the lettuce S16 treatment and the control, of which 724 were upregulated and 491 were downregulated (Fig 1B). There were 1272 DEGs between the lettuce S24 treatment and the control, of which 825 were upregulated and 447 were downregulated (Fig 1C). The number of upregulated genes was significantly higher than the number of downregulated genes under $CO_2$ enrichment in the three lettuce cultivars. There were 643, 654 and 695 unique DEGs among SCD6_vs_SCF6, SCDB16_vs_SCFB16 and SCD24_vs_SCF24, respectively. In addition,

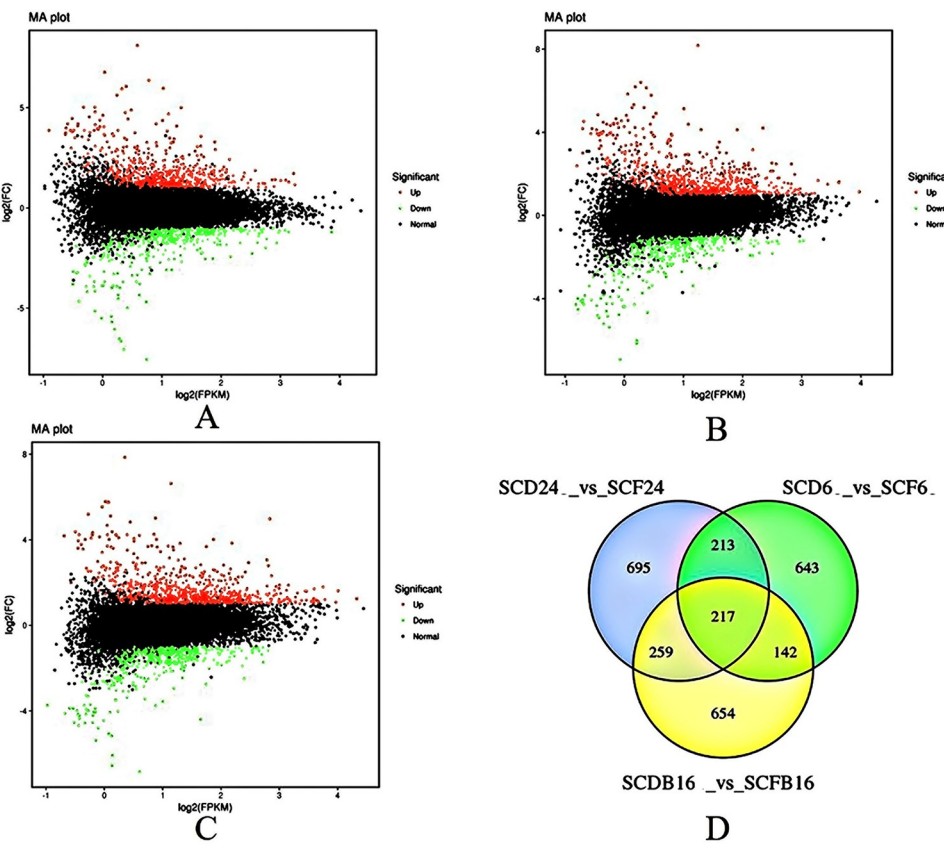

**Fig 1. Statistical analysis of DEGs under $CO_2$ enrichment in S6, S16 and S24.** (A) MA diagram of differential genes in S6. (B) MA diagram of differential genes in S16. (C) MA diagram of differential genes in S24. (D) Venn diagram of differential genes between carbon-rich group and the control group in the S6, S16 and S24.

there were 217 common DEGs between the three comparison groups (Fig 1D), among which 166 genes were upregulated and 44 genes were downregulated in the carbon-rich environment. Additionally, the expression trends of 7 genes (*Lsat_1_v5_gn_1_58781*, *Lsat_1_v5_gn_7_76640*, *Lsat_1_v5_gn_7_27480*, *Lsat_1_v5_gn_6_7601*, *Lsat_1_v5_gn_3_21161*, *Lsat_1_v5_gn_3_81620*, *Lsat_1_v5_gn_2_97741*, etc.) were inconsistent in the three materials.

**3.4.3 Enrichment KEGG analysis responding to elevated $CO_2$.** KEGG pathway analysis showed that the 217 DEGs were annotated into 34 pathways (S4 Table), which were related mainly to one cell process (transport and catabolism), one environmental information processing (plant hormone signal transduction), two genetic information processes (RNA decomposition and endoplasmic reticulum protein processing), 29 metabolic processes (photosynthesis, nitrogen metabolism, amino acid metabolism, glucose metabolism, chlorophyll metabolism, etc.), one biological system (circadian rhythm plants) and other biological systems. Significant enrichment analysis was performed on 217 DEGs by KEGG pathways; among these pathways, there were 8 pathways with a *P* value≤0.05, including limonene and pinene degradation (KO00903), starch and sucrose metabolism (KO00500), plant hormone signal transduction (KO04075), biosynthesis of unsaturated fatty acids (KO01040), metabolism of galactose (KO00052), biosynthesis of stilbenes, stilbenoid, diarylheptanoid and gingerol biosynthesis (KO00945), regulation of autophagy (KO04140), and pentose and glucuronate interconversions (KO00040).

## 3.5 Screening related genes responding to elevated $CO_2$

**3.5.1 Screening related enzyme genes involved in the responses to elevated $CO_2$.** Based on GO and KEGG enrichment analyses, a total of 12 key structural genes were highly correlated with $CO_2$ enrichment (Fig 2A, S5 Table). Including ethylene receptor gene *ETR* (*Lsat_1_v5_gn_3_122401* and *Lsat_1_v5_gn_8_164760*), *EBF1/2* gene (*Lsat_1_v5_gn_5_95460* and *Lsat_1_v5_gn_7_45101*) encoding F-box protein, Encoding of uroporphyrinogen decarboxylase (EC: 4.1.1.37) gene *Lsat_1_v5_gn_7_73881* during porphyrin and chlorophyll metabolism, β-furan fruit glycosidase (EC: 3.2.1.26) encoding gene (EC: 3.2.1.21) *Lsat_1_v5_gn_3_3061*, trehalose 6-phosphate synthetase gene *Lsat_1_v5_gn_4_142020*, hexokinase (EC: 2.7.1.1) encoding gene *Lsat_1_v5_gn_6_26160*, β-glucosidase (EC: 3.2.1.21) encoding gene *Lsat_1_v5_gn_9_23900*, and carboxylic anhydrase encoding gene *Lsat_1_v5_gn_4_182521*, etc., were upregulated. However, the jasmone ZIM domain-containing protein-encoding gene *Lsat_1_v5_gn_5_139561* and the β-amylase (EC: 3.2.1.2)-encoding gene *Lsat_1_v5_gn_3_15861* were downregulated. Among these genes, the transcriptional levels of 6 randomly selected genes were verified by qRT-PCR analysis, including *Lsat_1_v5_gn_5_139561*, *Lsat_1_v5_gn_6_26160*, *Lsat_1_v5_gn_7_73881*, *Lsat_1_v5_gn_9_23900*, *Lsat_1_v5_gn_4_142020*, and *Lsat_1_v5_gn_4_182521*. (Fig 3A–3F).

**3.5.2 Screening and identification of transcription factors involved in the responses to elevated $CO_2$.** In addition to genes encoding key enzymes of metabolic pathways, transcription factors that respond to high $CO_2$ concentrations also play an important role in plant growth and development and nutritional quality improvement. Thirty-one candidate transcription factors with high correlation were screened from 217 DEGs (Fig 2B, S5 Table), of which 25 genes were upregulated and 6 genes were downregulated under elevated $CO_2$. By family division, nine *AP2* transcription factors (*Lsat_1_v5_gn_5_108800*, *Lsat_1_v5_gn_2_128021*, *Lsat_1_v5_gn_1_5340*. etc.) and two auxin-binding proteins (*Lsat_1_v5_gn_5_136060* and *Lsat_1_v5_gn_5_141001*) were related to the hormone response; two bZIP transcription factors (*Lsat_1_v5_gn_9_59881* and *Lsat_1_v5_gn_7_102900*), three GATA transcription factors (*Lsat_1_v5_gn_3_128781*, *Lsat_1_v5_gn_1_34581* and *Lsat_1_v5_gn_7_37240*) and two

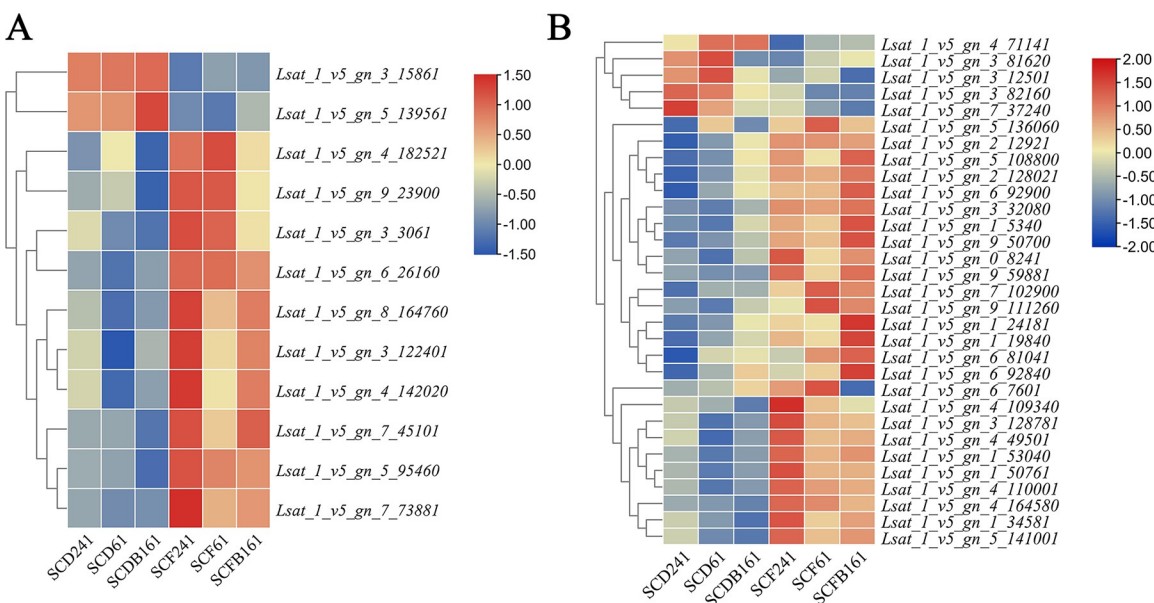

**Fig 2. Expression profiles of DGEs under $CO_2$ enrichment.** A and B respectively indicated structural genes and transcription factors responding to $CO_2$ enrichment.

zinc finger structure transcription factors (*Lsat_1_v5_gn_1_24181* and *Lsat_1_v5_gn_6_7601*) were associated with the light response; 2 *BHLH* transcription factors (*Lsat_1_v5_gn_4_49501 and Lsat_1_v5_gn_4_110001*), 8 *MYB* transcription factors (*Lsat_1_v5_gn_1_50761*, *Lsat_1_v5_gn_3_12501*, *Lsat_1_v5_gn_0_8241*. etc.), two chloroplast-related transcription factors (*Lsat_1_v5_gn_4_109340* and *Lsat_1_v5_gn_2_12921*) and one *NAC* transcription factor (*Lsat_1_v5_gn_4_71141*). The transcription levels of *Lsat_1_v5_gn_5_141001* and *Lsat_1_v5_gn_5_136060* had been verified by qRT-PCR analysis (Fig 3G and 3H).

The above results suggested that $CO_2$ enrichment can affect the transcription levels of structural genes and related transcription factors in various regulatory pathways, such as hormone

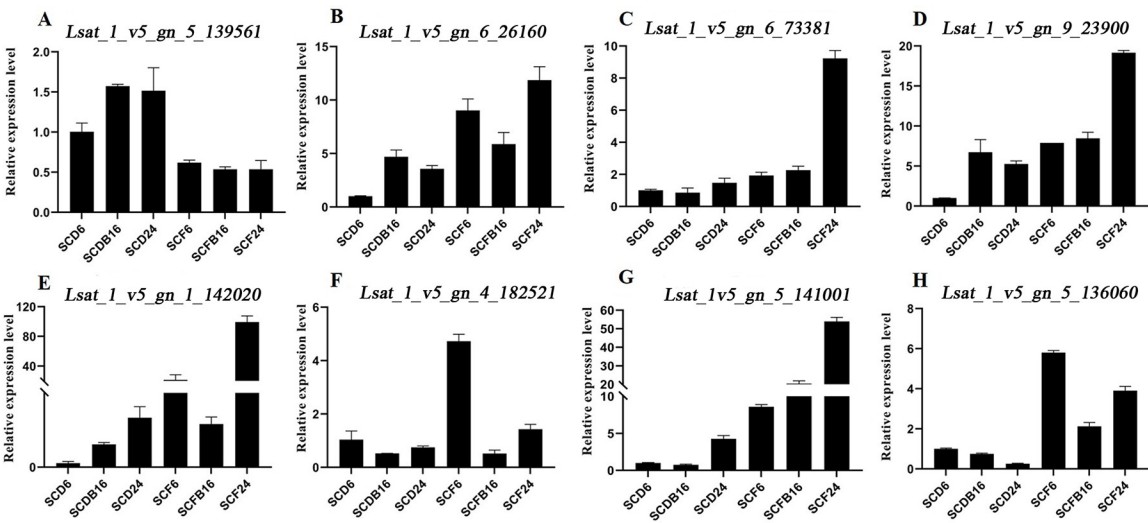

**Fig 3. qRT-PCR validation of DGEs results based on gene expression levels.**

induction, photosynthesis, nitrogen metabolism, sugar metabolism and amino acid metabolism, of lettuce, thus leading to changes in the types and content of downstream metabolites. which ultimately affect the morphological characteristics and nutritional quality of lettuce.

### 3.6 qRT–PCR verification

The transcription levels of eight genes, namely, *Lsat_1_v5_gn_7_73881* (uroporphyrinogen decarboxylase), *Lsat_1_v5_gn_5_13956*1 (jasmonate JAZ protein) and *Lsat_1_v5_gn_5_136060* (auxin-binding protein ABP), were detected by qRT-PCR (Fig 3). In the three lettuce varieties, the transcription levels of these genes were changed under elevated $CO_2$, except for *Lsat_1_v5_gn_5_139561*, and the expression of other genes was upregulated, which was consistent with the expression trend in the transcriptome data. These results provide important evidence for analyzing the molecular mechanism of crop response to elevated $CO_2$ and improving crop quality and yield.

## 4. Discussion

### 4.1 Effects of $CO_2$ enrichment on the growth and quality of lettuce

The results showed that elevated $CO_2$ promoted the morphological development and plant growth of lettuce. Diao et al. [17] found that $CO_2$ application could improve the plant height, plant weight and yield of leaf-use lettuce in greenhouses. Elevated $CO_2$ in the environment will lead to changes in the content of various metabolites in plants, which will further affect their nutritional quality [18]. A review showed that elevated $CO_2$ increased the concentrations of fructose, glucose, total soluble sugar, total antioxidant capacity, total phenols, total flavonoids and ascorbic acid in the edible part of vegetables but decreased the concentrations of nitrate [19]. In this study, elevated $CO_2$ significantly increased the content of vitamin C and chlorophyll in lettuce and reduced the accumulation of nitrate nitrogen, basically consistent with previous research results showing that elevated $CO_2$ can promote the accumulation of nutrients in lettuce. This result fully indicates that an appropriate concentration of elevated $CO_2$ can promote the growth of lettuce plants and improve nutritional quality.

Under elevated $CO_2$, the total chlorophyll content of Pak chio increased [20]. The total chlorophyll content also increased, but the chlorophyll b content decreased in soybean under drought stress with $CO_2$ enrichment [21]. The reason remains to be further studied. In addition, the latest research [22, 23] shows that, appropriate amount of nitrogen fertilizer can promote the utilization of $CO_2$ gas fertilizer, which points out the direction for the efficient agricultural production and rational fertilization.

### 4.2 Effects of $CO_2$ enrichment on photosynthesis

For most plants, changes in $CO_2$ concentration are the main factor affecting photosynthesis. Under high $CO_2$ concentrations, the total photosynthetic rate of ginger seedlings increased by 69% [24]. Under a carbon-rich environment, the net photosynthetic rate of pepper increased significantly, the LSP increased, and the LCP decreased simultaneously [25]. In this study, the maximum photosynthetic net increased, LSP significantly increased, and LCP significantly decreased, which partly explained the reasons for promoting growth.

Photoresponsive *GATA* and *bZIP* transcription factors are widely present in horticultural plants. *GATA* plays an important regulatory role in plant photosynthesis, chlorophyll metabolism and synthesis, carbon and nitrogen metabolism and other biological processes [26]. *bZIP* transcription factors play an important role in plant growth and development, light response and various forms of resistance to adversity stress [27, 28]. Two *bZIP* transcription factors

(*Lsat_1_v5_gn_9_59881* and *Lsat_1_v5_gn_7_102900*) and three *GATA* transcription factors (*Lsat_1_v5_gn_3_128781*, *Lsat_1_v5_gn_1_34581*, *Lsat_1_v5_gn_7_37240*) were screened. Of these five transcription factors, only *Lsat_1_v5_gn_7_37240* was down-regulated. *Lsat_1_v5_gn_4_109340* encodes the photosynthetic antenna protein, and *Lsat_1_v5_gn_2_12921* is the chloroplast precursor. *Lsat_1_v5_gn_7_73881* encodes uroporphyrinogen decarboxylase (EC: 4.1.1.37), which is a key enzyme in plant chlorophyll, phytochrome and heme synthesis [29]. *Lsat_1_v5_gn_4_182521* encodes carbonic anhydrase, which can accelerate the diffusion of inorganic carbon to the active site of carboxylase and increase the fixed rate of $CO_2$ by increasing the concentration of inorganic carbon around carboxylase [30]. Among these four genes, only *Lsat_1_v5_gn_4_109340* was expressed inconsistently among the three varieties, and other genes were up-regulated in all varieties under $CO_2$ enrichment conditions. The comprehensive analysis showed that the application of $CO_2$ improved the light energy utilization rate to different extents and caused the plants to accumulate more photosynthetic products, which may partially explain the acceleration of leaf growth under $CO_2$ enrichment of lettuce. As a C 3 plant, lettuce also showed strong sensitivity to high $CO_2$ concentrations.

## 4.3 Effects of $CO_2$ enrichment on carbohydrate metabolism

The main organic substances produced by photosynthesis are sugars, including monosaccharides, polysaccharides and starches, and starch is the most common. β-Amylase (EC3.2.1.2) is found mainly in higher plants and can decompose amylose into maltose [31]. In this study, the expression of *Lsat_1_v5_gn_3_15861* encoding β-amylase was downregulated, indicating that the degree of starch hydrolysis in lettuce was gradually weakened after $CO_2$ application, which was conducive to preventing the decrease in starch content in the plant. β-Glucosidase (EC:3.2.1.21) is a glucosidase that hydrolyzes mainly glycosidic bonds. Under elevated $CO_2$, the expression of β-glucosidase *Lsat_1_v5_gn_9_23900* in lettuce was upregulated, and the activity of the enzyme was enhanced, which promoted the transformation of fibrous sugar to β-D-glucose, which was beneficial to the synthesis of glucose. β-Furan fruit glycosidase (EC3.2.1.26) plays a very important role in sucrose metabolism. When β-furan fruit glycosidase activity is high, sucrose hydrolyzes, thereby inhibiting sucrose accumulation. The expression of *Lsat_1_v5_gn_3_3061*, which encodes β-furan glucosidase, in lettuce was upregulated after $CO_2$ application, which indicated that β-furan fruit glycosidase activity was enhanced, and sucrose decomposition was accelerated.

Combined with the expression analysis of all the related genes in the starch and sucrose metabolic pathways, elevated $CO_2$ was found to accelerate the degradation of sucrose, while the accumulation of starch and various types of sugars increased, which provided a large amount of substrate for the chloroplast to decompose starch at night and provided energy material for the growth and development of lettuce.

## 4.4 Effects of $CO_2$ enrichment on substance metabolism in plant growth

Plant growth substances contain plant hormones and plant growth regulators, which have significant regulatory effects on plant growth and development. Studies have shown that the ethylene signaling pathway can inhibit the ethylene response in plants by activating the ethylene receptor protein ETR-encoded gene, preventing the overactivated ethylene signaling pathway from having significant inhibitory and toxic effects on plant growth [32]. Both *Lsat_1_v5_gn_3_122401* and *Lsat_1_v5_gn_8_164760*, which encode ETR receptor proteins, were upregulated in lettuce, which may further enhance ETR receptor activity and bind with downstream CTR1 proteins to interrupt ethylene signal transmission and inhibit the ethylene response. At the same time, the ethylene response in the ethylene signaling pathway is also

regulated by two F-box proteins (EBF1/2) [33]. Under elevated CO$_2$, *Lsat_1_v5_gn_5_95460* and *Lsat_1_v5_gn_7_45101*, encoding the EBF1/2 protein, were upregulated; that is, the EBF1/2 protein increased, which also inhibited the ethylene response.

Ethylene response factor (*AP2-ERF*) is a class of widely existing transcription factors in plants that play an important role in the regulation of plant growth and development, secondary metabolic accumulation and the ability to resist adversity [34–36]. A total of 9 *AP$_2$-ERF* transcription factors were screened in this study, and all of these transcription factors were upregulated. The increase in their transcription level may make lettuce grow faster, accumulate more secondary metabolites and have stronger stress resistance in a high CO$_2$ concentration environment.

Auxins synergistically regulate cell division and cell expansion and control shoot meristem development and stem elongation [37]. *Lsat_1_v5_gn_5_136060* and *Lsat_1_v5_gn_5_141001* encode auxin-binding proteins, both of which function in nutrient depot activity, and *Lsat_1_v5_gn_5_141001* is also involved in photosynthesis.

Jasmone ZIM domain-containing protein can regulate the release of downstream transcription factors or signaling proteins, thereby promoting various growth and development processes and antistress responses regulated by jasmine [38]. The expression of *Lsat_1_v5_gn_5_139561*, which encodes JAZ protein, in lettuce was downregulated under the induction of high concentrations of CO$_2$, and JAZ protein could not form dimers and could not inhibit the jasmone response pathway in plants to promote the regulation of jasmine on plant growth and development and improve plant resistance.

Simultaneous application of CO$_2$ under drought conditions, all genes related to the regulation of metabolic pathways such as ETH, SA, JA, and IAA are up-regulated in soybean [39], which is not completely consistent with the gene expression law in this study. But both indicate that the complex tandem network between hormones is an important way for plants to adjust their response to external stimuli. But they all show that the complex tandem network between hormones is an important way for plants to adjust to external stimuli.

### 4.5 Effects of CO$_2$ enrichment on *MYB* transcription factors

As one of the largest transcription factor families in plants, *MYB* transcription factors are widely involved in regulating plant growth and development, secondary metabolite synthesis and resistance to various abiotic stresses [40, 41]. In this study, a total of 8 *MYB* transcription factors were screened from DEGs. *Lsat_1_v5_gn_3_12501*, *Lsat_1_v5_gn_1_53040*, *Lsat_1_v5_gn_3_81620*, and *Lsat_1_v5_gn_9_111260* are homologous genes of *MYB9A*, *MYB1*, *MYB44* and *MYB44*, respectively. These transcription factors may play an important regulatory role in the growth and development of lettuce and its ability to resist adversity in carbon-rich environments [42, 43] *Lsat_1_v5_gn_3_82160* and *Lsat_1_v5_gn_1_50761* are homologous genes of *MB10* and *SANT*, which may play an important regulatory role in anthocyanin accumulation [44]. The expression of *Lsat_1_v5_gn_3_82160* was downregulated in the three materials in the CO$_2$ treatment area, and *Lsat_1_v5_gn_1_50761* was upregulated. Whether a high CO$_2$ concentration can promote or inhibit the accumulation of anthocyanin in lettuce and how to regulate the accumulation of anthocyanin content in a high CO$_2$ environment still need further study.

### 5. Conclusions

The present study showed that elevated CO$_2$ promoted the growth of lettuce leaves, increased photosynthetic efficiency, and improved quality compared to ambient conditions. Analysis of transcriptomes using RNA-seq revealed 13 structural genes and 31 transcription factors

involved in the response to $CO_2$ enrichment. The results showed that $CO_2$ enrichment was involved in the light response, promoted the accumulation of chlorophyll, starch and sucrose, regulated ethylene, auxins and jasmonic acid signals, and finally stimulated the growth and nutritional quality of lettuce. These research results provide a theoretical reference for the high-yield and high-quality cultivation of lettuce in solar greenhouses and the application of $CO_2$ fertilization technology.

## Supporting information

**S1 Fig. Materials for testing (three materials respectively represent 3 colors, S6 represents green, S16 represents green and purple, and S24 represents purple lettuce).**
(PDF)

**S1 Table. The primer sequences for qRT-PCR.**
(PDF)

**S2 Table. Sample sequencing data evaluation statistics.**
(PDF)

**S3 Table. Functional annotation of new genes in *Lactuca sativa*.**
(XLSX)

**S4 Table. KEGG significant enrichment of different expression genes.**
(PDF)

**S5 Table. $CO_2$ response-related differentially expressed genes.**
(XLSX)

## Author Contributions

**Conceptualization:** Hongxia Song, Meilan Li, Xiaoyong Xu.

**Data curation:** Xiaonan Lu, Tianyue Song, Qiang Lu.

**Formal analysis:** Hongxia Song, Peiqi Wu.

**Investigation:** Hongxia Song, Peiqi Wu, Bei Wang.

**Resources:** Xiaonan Lu, Xiaoyong Xu.

**Writing – original draft:** Hongxia Song, Meilan Li, Xiaoyong Xu.

**Writing – review & editing:** Hongxia Song, Peiqi Wu, Xiaonan Lu, Bei Wang, Tianyue Song, Qiang Lu, Meilan Li, Xiaoyong Xu.

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
