## [Decision Letter · Decision Letter 0]

15 Jun 2022

PONE-D-22-09910Comparative physiological and transcriptomic analyses reveal the mechanisms of CO2 enrichment in promoting the growth and quality in Lactuca sativaPLOS ONE

Dear Dr. Song,

Thank you for submitting your manuscript to PLOS ONE. After careful consideration, we feel that it has merit but does not fully meet PLOS ONE’s publication criteria as it currently stands. Therefore, we invite you to submit a revised version of the manuscript that addresses the points raised during the review process.

Please note that we have only been able to secure a single reviewer to assess your manuscript. We are issuing a decision on your manuscript at this point to prevent further delays in the evaluation of your manuscript. Please be aware that the editor who handles your revised manuscript might find it necessary to invite additional reviewers to assess this work once the revised manuscript is submitted. However, we will aim to proceed on the basis of this single review if possible.  Your manuscript has been assessed by an expert reviewer, whose comments are appended below. The reviewer has highlighted concerns about several methodological details which are missing, among other issues. Please ensure you respond to each point carefully in your response to reviewers document, and modify your manuscript accordingly.

We look forward to receiving your revised manuscript.

Kind regards,

Joseph Donlan

Editorial Office

PLOS ONE

Journal Requirements:

 “201703D211001-04-01, 201903D211011

Shanxi Province Key Research and Development Program Key Project analysis”

“This work was supported by the Shanxi Province Key Research and Development Program Key Projects under Grant (201703D211001-04- 01, 201903D211011).”

“201703D211001-04-01, 201903D211011

Shanxi Province Key Research and Development Program Key Project analysis”

Reviewers' comments:

Reviewer's Responses to Questions

**Comments to the Author**

1. Is the manuscript technically sound, and do the data support the conclusions?

Reviewer #1: Partly

2. Has the statistical analysis been performed appropriately and rigorously? 

Reviewer #1: No

3. Have the authors made all data underlying the findings in their manuscript fully available?

Reviewer #1: No

4. Is the manuscript presented in an intelligible fashion and written in standard English?

Reviewer #1: Yes

5. Review Comments to the Author

Reviewer #1: The manuscript written by Song et al. PONE-D-22-09910 `Comparative physiological and transcriptomic analyses reveal the mechanisms of CO2 enrichment in promoting the growth and quality in Lactuca sativa` investigates the effects of CO2 enrichment on physiology, growth and transcripts of Lactuca sativa. Here, authors investigate the CO2 effects inside a greenhouse, under controlled conditions. In the context of current climate changes, CO2 studies are crucial for a better understanding of future food security and productivity. The manuscript is well-written, clear, and experiments were conducted with adequate methods. However, much more details are necessary in the methods section. Therefore, the manuscript can be accepted after major revisions. A list with my suggestions can be found bellow:

Title

Title is clear and informative.

Abstract

In general, abstract is well-written and concise. I listed a few recommendations bellow:

Page 2. in `… attention from all walks of life…`the expression walks of life seems strange here, maybe authors can rewrite this sentence in a more scientific approach.

Regarding the sentence ` The rational utilization of CO2 by plants is an effective way to promote carbon neutrality`. How is possible that lettuce cultivation will promote carbon neutrality? Carbon neutrality achievement by plants is only possible when large carbon flux is occurring, such as CO2 flux in forest and other ecosystems. This information seems out of context here. In addition, authors need to decide what will be focus of the manuscript: (1) anthropogenic increase of atmospheric CO2 or (2) CO2 fertilization techniques in controlled environments to increase lettuce production. In the situation 1, authors need to put into account that productivity of plants in the future will also occur under elevated temperatures conditions and different rainfall patterns, therefore, the results obtained here cannot be extrapolated to future conditions since both, temperature and drought can negate or fully mitigate the benefits of a CO2 increase. In the situation 2, authors can focus only in CO2 fertilization under controlled conditions for a higher lettuce production, without taking into account the future increases in temperature, since plants are growing under controlled conditions (controlled temperature, CO2, etc). It seems that situation 2 is the more adequate approach for this manuscript and all sentences related to climate change and atmospheric CO2 increased by human activity are not necessary and should be removed.

Along the manuscript authors used sentences such as `… effect of CO2 on lettuce…`. Authors are not testing the effects of CO2 on plants, but the effects of ELEVATED CO2 on plants. In phrases such as this one, it is important to include this information.

Introduction

In general, introduction is well-written and concise. However, the introduction needs a hypothesis. Which plant responses authors expected according to the literature? In the discussion section, authors should clarify if the hypothesis was corroborated or not.

Material and methods

S1 Fig. Please, insert a subtitle in the supplementary figure.

Why authors decided to initiate the CO2 enrichment only after one week of transplanting? Is there any reason for this?

Please, change `release time` for `CO2 fumigation period `

Why CO2 was not fumigated in rainy or snowy days? How many rainy/snowy days occurred during the experiment? Please, include both information in the manuscript.

Please, describe in more details what `traditional management practices` means. I imagine that each lettuce color has a different management strategy. In addition, in different countries `traditional management practices` for lettuce cultivation will have different meanings.

Which gas exchange parameters were measured? How light curves (light intensity steps) were conducted to obtain LSP and LCP parameters?

Please, include the statistical methods used to analyze the data in the entire manuscript in the material and methods section. This information is crucial for data interpretation. Did authors used ANOVA? Student t tests?

Results

Within each table we have lowercase and uppercase letters presumably indicating statistical differences between average values. However, we can`t find the meaning of these letters in the subtitle or methods section. Please, include this information in all subtitles.

Page 10. The following sentence `The comprehensive analysis showed that the application of CO2 improved the light energy utilization rate to different extents and caused the plants to accumulate more photosynthetic products, which may partially explain the acceleration of leaf growth under CO2 enrichment of lettuce.` should be placed in the discussion section.

Page 11. Again, sentences such as `Elevated CO2 in the environment will lead to changes in the content of various metabolites in plants, which will further affect their nutritional quality [16].` should be placed in other sections of the manuscript such as introduction or discussion.

All subtitles can be improved with more details. In the current form, subtitles are too simple.

Discussion

It was a great idea to split the discussion in sub-sections. In general, discussion section is ok. However, especially in the subsection of photosynthesis, the discussion is too simple. Much more mechanistic discussion is possible using your data.

6. PLOS authors have the option to publish the peer review history of their article (what does this mean?). If published, this will include your full peer review and any attached files.

Reviewer #1: No

---

## [Author Response · Author response to Decision Letter 0]

1 Aug 2022

Response to Reviewer 1:

Abstract

1. Response to comment: (Page 2. in`… attention from all walks of …`the expression walks of life seems strange here, maybe authors can rewrite this sentence in a more scientific approach.)

Response: ‘from all walks of life’ had been deleted.

2. Response to comment: (Regarding the sentence `The rational utilization of CO2 by plants is an effective way to promote carbon neutrality`. How is possible that lettuce cultivation will promote carbon neutrality? Carbon neutrality achievement by plants is only possible when large carbon flux is occurring, such as CO2 flux in forest and other ecosystems. This information seems out of context here. In addition, authors need to decide what will be focus of the manuscript: (1) anthropogenic increase of atmospheric CO2 or (2) CO2 fertilization techniques in controlled environments to increase lettuce production. In the situation 1, authors need to put into account that productivity of plants in the future will also occur under elevated temperatures conditions and different rainfall patterns, therefore, the results obtained here cannot be extrapolated to future conditions since both, temperature and drought can negate or fully mitigate the benefits of a CO2 increase. In the situation 2, authors can focus only in CO2 fertilization under controlled conditions for a higher lettuce production, without taking into account the future increases in temperature, since plants are growing under controlled conditions (controlled temperature, CO2, etc). It seems that situation 2 is the more adequate approach for this manuscript and all sentences related to climate change and atmospheric CO2 increased by human activity are not necessary and should be removed.

Response: All sentences related to climate change and atmospheric CO2 increased by human activity had been removed.

3. Response to comment: (Along the manuscript authors used sentences such as `… effect of CO2 on lettuce…`. Authors are not testing the effects of CO2 on plants, but the effects of ELEVATED CO2 on plants. In phrases such as this one, it is important to include this information.)

Response: Full text revised as requested.

Introduction 

1. Response to comment: (In general, introduction is well-written and concise. However, the introduction needs a hypothesis. Which plant responses authors expected according to the literature? In the discussion section, authors should clarify if the hypothesis was corroborated or not.) 

Response: This part has been added in the Introduction and Discussion respectively.

Material and methods

1. Response to comment: (S1 Fig. Please, insert a subtitle in the supplementary figure.)

Response: S1 Fig. had been inserted a subtitle in the supplementary figure.

2. Response to comment: (Why authors decided to initiate the CO2 enrichment only after one week of transplanting? Is there any reason for this?)

Response: Plants could completely grow normally one week after transplanting, and rejuvenation period was ended.

3. Response to comment: (Please, change `release time` for `CO2 fumigation period`)

Response: `release time` had been revised.

4. Response to comment: (Why CO2 was not fumigated in rainy or snowy days? How many rainy/snowy days occurred during the experiment? Please, include both information in the manuscript.)

Response: In rainy and snowy days, the photosynthetic rate decreases, and higher concentrations of CO2 cannot be absorbed by plants. and there were 30 d for CO2 application in general. These are also included in the Materials and methods section.

5. Response to comment: (Please, describe in more details what `traditional management practices` means. I imagine that each lettuce color has a different management strategy. In addition, in different countries `traditional management practices` for lettuce cultivation will have different meanings.)

Response: This experiment adopts traditional soil cultivation, which is supplemented in the text. The specific management steps are cited in the literature [12].

6. Response to comment: (Which gas exchange parameters were measured? How light curves (light intensity steps) were conducted to obtain LSP and LCP parameters?)

Response: The measured gas exchange parameters include Gs and Ci. One leaf per plant was chosen for conducting the light curve, and the light intensity settings were 50, 100, 150, 200, 300, 400, 600, 800, 1000, 1200, 1400, 1600, 1800, 2000 and 2200 μmol•m−2•s−1, respectively. Related content has been supplemented in the Materials and methods section.

7. Response to comment: (Please, include the statistical methods used to analyze the data in the entire manuscript in the material and methods section. This information is crucial for data interpretation. Did authors used ANOVA? Student t tests?)

Response: Statistical methods had been supplemented.

Results

1. Response to comment: (Within each table we have lowercase and uppercase letters presumably indicating statistical differences between average values. However, we can`t find the meaning of these letters in the subtitle or methods section. Please, include this information in all subtitles.)

Response: The meaning of lowercase and uppercase letters had been supplemented.

2. Response to comment: (Page 10. The following sentence `The comprehensive analysis showed that the application of CO2 improved the light energy utilization rate to different extents and caused the plants to accumulate more photosynthetic products, which may partially explain the acceleration of leaf growth under CO2 enrichment of lettuce.` should be placed in the discussion section.) 

Response: This sentence had been placed in the photosynthetic part of the discussion.

3. Response to comment: (Page 11. Again, sentences such as `Elevated CO2 in the environment will lead to changes in the content of various metabolites in plants, which will further affect their nutritional quality [16].` should be placed in other sections of the manuscript such as introduction or discussion.) 

Response: This sentence has been placed in the Lettuce Growth and Quality section of the discussion.

4. Response to comment: (All subtitles can be improved with more details. In the current form, subtitles are too simple.)

Response: All subtitles are modified one by one.

Discussion

1. Response to comment: (It was a great idea to split the discussion in sub-sections. In general, discussion section is ok. However, especially in the subsection of photosynthesis, the discussion is too simple. Much more mechanistic discussion is possible using your data.)

Response: Discussion section had been added.

---

## [Decision Letter · Decision Letter 1]

19 Sep 2022

PONE-D-22-09910R1Comparative physiological and transcriptomic analyses reveal the mechanisms of CO2 enrichment in promoting the growth and quality in Lactuca sativaPLOS ONE

Dear Dr. Song,

Thank you for submitting your manuscript to PLOS ONE. After careful consideration, we feel that it has merit but does not fully meet PLOS ONE’s publication criteria as it currently stands. Therefore, we invite you to submit a revised version of the manuscript that addresses the points raised during the review process.

We look forward to receiving your revised manuscript.

Kind regards,

Mohammad Irfan, Ph.D.

Academic Editor

PLOS ONE

Additional Editor Comments:

In particular, I agree with reviewer #3 who has valid concerns. 

Reviewers' comments:

Reviewer's Responses to Questions

**Comments to the Author**

1. If the authors have adequately addressed your comments raised in a previous round of review and you feel that this manuscript is now acceptable for publication, you may indicate that here to bypass the “Comments to the Author” section, enter your conflict of interest statement in the “Confidential to Editor” section, and submit your "Accept" recommendation.

Reviewer #2: All comments have been addressed

Reviewer #3: (No Response)

2. Is the manuscript technically sound, and do the data support the conclusions?

Reviewer #2: Yes

Reviewer #3: Partly

3. Has the statistical analysis been performed appropriately and rigorously? 

Reviewer #2: Yes

Reviewer #3: No

4. Have the authors made all data underlying the findings in their manuscript fully available?

Reviewer #2: Yes

Reviewer #3: No

5. Is the manuscript presented in an intelligible fashion and written in standard English?

Reviewer #2: Yes

Reviewer #3: Yes

6. Review Comments to the Author

Reviewer #2: The paper describes about “Comparative physiological and transcriptomic analyses reveal the mechanisms of CO2 enrichment in promoting the growth and quality in Lactuca sativa”

Authors have revised but still new references are missing

Yr 2022- nil

Yr 2021~1

So, include relevant references to update discussion

Reviewer #3: The manuscript titled “Comparative physiological and transcriptomic analyses reveal the mechanisms of CO2 enrichment in promoting the growth and quality in Lactuca sativa” investigated the effects of elevated CO2 on three varieties of lettuce. The author found that elevated CO2 promoted the photosynthesis of lettuce, as well as the growth and quality of lettuce. The authors also performed RNA-seq for the three varieties of lettuce under elevated CO2 and identified responsive genes (and their associated pathways) shared by all three varieties. Overall, the manuscript easy to follow. However, it is very descriptive and lacks necessary experimental details. Please see detailed comments below.

Selected major comments are listed below:

1. Experiment details about CO2 application, what is the concentration of ambient CO2 and what is the concentration of elevated CO2, was the CO2 concentration monitored throughout the experiment?

2. Please clarify how the statistical significance was obtained, was the comparison between elevated CO2 and ambient CO2 done for each variable (from plant growth, photosynthesis, lettuce quality) and each variety or there are different grouping?

3. Line 115-116 and 118-119: “One leaf per plant was chosen for conducting the light curve” and “Three healthy leaves were selected for each measurement”. Please provide details for how these leaf were selected, were they consistent between different plants?

4. Line 133-136: It seems that there is only one replicate used for RNA-seq from S6, S16, S24 varieties. Please clarify this in the methods.

5. Line 140-147: Please provide additional information for 1) how the RNA-seq data was pre-processed; 2) how the RNA-seq reads were mapped (tool and genome used); 3) how DEGs were identified (which tool was used); 4) whether DEGs were identified for each lettuce variety separately or all together; and 5) how functional enrichment was performed for both KEGG and GO (what tools were used). Reference Sun et al. doesn’t provide these details.

6. Line 177: 1) Please clarify how leaf width, plant height and width were defined in the current study; 2) Please reformat Table 1 to have a clear separation between S6, S16 and S24 groups.

7. The authors mentioned multiple RNA-seq studies from line 64-74 about the effect of elevated CO2 in different species. Did the authors compare the results from current study to the results from these previous studies to see if there are any genes/pathways being differentially regulated by elevated CO2 regardless of species or these species might have very distinct response to the elevated CO2?

Selected minor comments:

1. Please provide figures with higher resolution.

2. Line 198: Please justify why there are uppercase and lowercase significance letters. Do they represent different grouping?

3. Data used in the current study doesn’t seem to be accessible accession number PRJNA859388 (http://www.ncbi.nlm.nih.gov/bioproject/859388).

7. PLOS authors have the option to publish the peer review history of their article (what does this mean?). If published, this will include your full peer review and any attached files.

Reviewer #2: **Yes: **Dr. Saurabh Yadav,

Reviewer #3: No

---

## [Author Response · Author response to Decision Letter 1]

27 Sep 2022

Dear Editor and Reviewers:

Thank you for your comments and for the reviewers’ comments concerning our manuscript entitled “Comparative physiological and transcriptomic analyses reveal the mechanisms of CO2 enrichment in promoting the growth and quality in Lactuca sativa”. Those comments are all valuable and very helpful for revising and improving our paper. We have checked the manuscript and revised it according to the comments. Changes were marked in colored text in the paper, Increased content was marked in red. Detailed responses to the comments of reviewer were attached. I hope that the revised manuscript is now suitable for publication.

Once again, thank you very much for your comments and suggestion.

Sincerely,

Hongxia Song

Response to Reviewer 2:

Response to comment: (Authors have revised but still new references are missing Yr 2022- nil Yr 2021~1)

Response: The discussion section had added content, using a total of 5 new references.

Response to Reviewer 3:

1. Response to comment: (Experiment details about CO2 application, what is the concentration of ambient CO2 and what is the concentration of elevated CO2?)

Response: The concentration of ambient CO2 is 400 μmol•mol-1 and the concentration of elevated CO2 is 800±50 μmol•mol-1, the CO2 concentration was monitored throughout the experiment. These were described in detail in lines 95-96 and lines 100-102.)

2. Response to comment: (Please clarify how the statistical significance was obtained, was the comparison between elevated CO2 and ambient CO2 done for each variable (from plant growth, photosynthesis, lettuce quality) and each variety or there are different grouping?)

Response: The comparison between elevated CO2 and ambient CO2 was done for the statistical significance for each variable (from plant growth, photosynthesis, lettuce quality) of each varietys. No significant difference analysis was performed between varieties. Relevant content has been provided and modified.

3. Response to comment: (Line 115-116 and 118-119: “One leaf per plant was chosen for conducting the light curve” and “Three healthy leaves were selected for each measurement”. Please provide details for how these leaf were selected, were they consistent between different plants?)

Response: A total of three plants were selected for each treatment of each variety, and one leaf was selected per plant for determination, and then the average value was calculated. The leaf positions of each variety and treatment were the same. Relevant content has been provided and modified.

4. Response to comment: (Line 133-136: It seems that there is only one replicate used for RNA-seq from S6, S16, S24 varieties. Please clarify this in the methods.) 

Response: There is only one replicate used for RNA-seq from S6, S16, S24 varieties. This had been clarified and supplemented in the methods.

5. Response to comment: (Line 140-147: Please provide additional information for 1) how the RNA-seq data was pre-processed; 2) how the RNA-seq reads were mapped (tool and genome used); 3) how DEGs were identified (which tool was used); 4) whether DEGs were identified for each lettuce variety separately or all together; and 5) how functional enrichment was performed for both KEGG and GO (what tools were used). Reference Sun et al. doesn’t provide these details.)

Response: All of additional information had been provided.

6. Response to comment: ( Line 177: 1) Please clarify how leaf width, plant height and width were defined in the current study; 2) Please reformat Table 1 to have a clear separation between S6, S16 and S24 groups.)

Response: Leaf width, plant height and width were defined in methods. Table 1 has been reformatted.

7. Response to comment: (The authors mentioned multiple RNA-seq studies from line 64-74 about the effect of elevated CO2 in different species. Did the authors compare the results from current study to the results from these previous studies to see if there are any genes/pathways being differentially regulated by elevated CO2 regardless of species or these species might have very distinct response to the elevated CO2?)

Response: C3 and C4 crops respond differently to CO2. Our team has been working on CO2 enrichment research of vegetable for many years, since most vegetables are C3 plants with short growth cycle and are sensitive to carbon dioxide, they all show promotion effect at present. This was also reflected in the discussion.

8. Response to comment: (Please provide figures with higher resolution.)

Response: The clarity of the figures has been improved.

9. Response to comment: (Line 198: Please justify why there are uppercase and lowercase significance letters. Do they represent different grouping?)

Response: Difference analysis was not performed between varieties, but significant difference analysis was performed only between controls and treatments of the same variety. These have been added in the note.

10. Response to comment: (Data used in the current study doesn’t seem to be accessible accession number PRJNA859388 (http://www.ncbi.nlm.nih.gov/bioproject/859388).

Response: The current link will not be released until the article is published, if necessary, you can use the following address to view. ( https://dataview.ncbi.nlm.nih.gov/object/PRJNA859388?reviewer=3q9f1u02qig63u5jeclng0tmai)

---

## [Decision Letter · Decision Letter 2]

20 Oct 2022

PONE-D-22-09910R2Comparative physiological and transcriptomic analyses reveal the mechanisms of CO2 enrichment in promoting the growth and quality in Lactuca sativaPLOS ONE

Dear Dr. Song,

Thank you for submitting your manuscript to PLOS ONE. After careful consideration, we feel that it has merit but does not fully meet PLOS ONE’s publication criteria as it currently stands. Therefore, we invite you to submit a revised version of the manuscript that addresses the points raised during the review process. Specifically, please address the comment raised by the reviewer with proper justification regarding the biological replicates. 

We look forward to receiving your revised manuscript.

Kind regards,

Mohammad Irfan, Ph.D.

Academic Editor

PLOS ONE

Journal Requirements:

Reviewers' comments:

Reviewer's Responses to Questions

**Comments to the Author**

1. If the authors have adequately addressed your comments raised in a previous round of review and you feel that this manuscript is now acceptable for publication, you may indicate that here to bypass the “Comments to the Author” section, enter your conflict of interest statement in the “Confidential to Editor” section, and submit your "Accept" recommendation.

Reviewer #3: (No Response)

2. Is the manuscript technically sound, and do the data support the conclusions?

Reviewer #3: Partly

3. Has the statistical analysis been performed appropriately and rigorously? 

Reviewer #3: Yes

4. Have the authors made all data underlying the findings in their manuscript fully available?

Reviewer #3: Yes

5. Is the manuscript presented in an intelligible fashion and written in standard English?

Reviewer #3: Yes

6. Review Comments to the Author

Reviewer #3: The authors addressed most of the initial comments. However, the use of only one replicate for RNA-seq is lower than the minimum requirement of three replicates, which will dramatically reduce the statistical power of detecting DEGs. One way the authors could consider is to use S6, S16, S24 varieties as biological replicates to investigate the effect of elevated CO2.

7. PLOS authors have the option to publish the peer review history of their article (what does this mean?). If published, this will include your full peer review and any attached files.

Reviewer #3: No

---

## [Author Response · Author response to Decision Letter 2]

28 Oct 2022

Dear Editor and Reviewers:

Thank you for your comments and for the reviewers’ comments concerning our manuscript entitled “Comparative physiological and transcriptomic analyses reveal the mechanisms of CO2 enrichment in promoting the growth and quality in Lactuca sativa”. Those comments are all valuable and very helpful for revising and improving our paper. We have checked the manuscript and revised it according to the comments. Increased content was marked in red. Detailed responses to the comments of reviewer were attached. I hope that the revised manuscript is now suitable for publication. 

The study was in part supported by the Shanxi Province Key Research and Development Program Key Projects under Grant (201703D211001-04-01), funded by the Shanxi Province Key Research and Development Program Key Projects under Grant (201903D211011). The funders had role in publication. 

Once again, thank you very much for your comments and suggestion.

Sincerely,

Hongxia Song

Response to Reviewer 3:

Response to comment: (The authors addressed most of the initial comments. However, the use of only one replicate for RNA-seq is lower than the minimum requirement of three replicates, which will dramatically reduce the statistical power of detecting DEGs. One way the authors could consider is to use S6, S16, S24 varieties as biological replicates to investigate the effect of elevated CO2.)

Thank you very much for your advice. In fact, we also studied three species（S6, S16, S24 varieties）as three biological replicates. Moreover, it has been supplemented in the manuscript.

---

## [Editor Report · Decision Letter 3]

11 Nov 2022

Comparative physiological and transcriptomic analyses reveal the mechanisms of CO2 enrichment in promoting the growth and quality in Lactuca sativa

PONE-D-22-09910R3

Dear Dr. Song,

We’re pleased to inform you that your manuscript has been judged scientifically suitable for publication and will be formally accepted for publication once it meets all outstanding technical requirements.

Kind regards,

Mohammad Irfan, Ph.D.

Academic Editor

PLOS ONE
---

## [Editor Report · Acceptance letter]

25 Jan 2023

PONE-D-22-09910R3 

Comparative physiological and transcriptomic analyses reveal the mechanisms of CO2 enrichment in promoting the growth and quality in *Lactuca sativa*

Dear Dr. Song:

I'm pleased to inform you that your manuscript has been deemed suitable for publication in PLOS ONE. Congratulations! Your manuscript is now with our production department. 

Kind regards, 

on behalf of

Dr. Mohammad Irfan 

Academic Editor

PLOS ONE